# Super Ductility of Nanoglass Aluminium Nitride

**DOI:** 10.3390/nano9111535

**Published:** 2019-10-29

**Authors:** Yinbo Zhao, Xianghe Peng, Cheng Huang, Bo Yang, Ning Hu, Mingchao Wang

**Affiliations:** 1College of Aerospace Engineering, Chongqing University, Chongqing 400044, China; zyb123@cqu.edu.cn (Y.Z.); huangcheng@cqu.edu.cn (C.H.); yangbo16@cqu.edu.cn (B.Y.); ninghu@cqu.edu.cn (N.H.); 2Department of Materials Science and Engineering, Faculty of Engineering, Monash University, Clayton, VIC 3800, Australia; Mingchao.Wang@monash.edu

**Keywords:** nano-glass ceramic, amorphous AlN, interface, ductility, molecular dynamics simulations

## Abstract

Ceramics have been widely used in many fields because of their distinctive properties, however, brittle fracture usually limits their application. To solve this problem, nanoglass ceramics were developed. In this article, we numerically investigated the mechanical properties of nanoglass aluminium nitride (ng-AlN) with different glassy grain sizes under tension using molecular dynamics simulations. It was found that ng-AlN exhibits super ductility and tends to deform uniformly without the formation of voids as the glassy grain size decreases to about 1 nm, which was attributed to a large number of uniformly distributed shear transformation zones (STZs). We further investigated the effects of temperature and strain rate on ng-AlN_*d* = 1 nm_, which showed that temperature insignificantly influences the elastic modulus, while the dependence of the ultimate strength on temperature follows the *T*^2/3^ scaling law. Meanwhile, the ultimate strength of ng-AlN_*d* = 1 nm_ is positively correlated with the strain rate, following a power function relationship.

## 1. Introduction

High-strength ceramics [1] have been extensively investigated owing to their high strength and hardness, and low density, etc. among which aluminium nitride (AlN), a member of nitrides with metals from the IIIA group has attracted considerable attention due to its distinctive properties, as well as its mechanical properties. For instance, AlN with a wurtzite structure can be used as an ideal radiator and electronic packaging material, due to its high thermal conductivity [2,3], low thermal expansion [4,5], and excellent insulation [6] properties. However, the tensile brittle rupture due to the strong covalent bonds makes their application strongly limited [7]. In order to overcome this deficiency, great efforts have been made to enhance its ductility and toughness. It was found by Heard et al. [8] that an intrinsic failure pattern of the AlN known as brittle fracture would be changed to plastic deformation under a high hydrostatic pressure condition. Meanwhile, it was found in an experiment that the decrease of the size of AlN pillar may induce brittle-to-ductile transition, due to the activation of dislocations [9]. Besides, at high pressure, metal oxides with an amorphous structure could undergo severe plastic deformation at moderate temperatures [10,11,12]. Zhao et al. [13] further investigated the deformation mechanism in amorphous AlN (a-AlN), and found that, compared with its crystalline counterpart, the enhanced ductility of a-AlN can be attributed to the self-repairing mechanism.

Nanoglass (NG) [14] is a class of novel non-crystalline materials, which was first proposed and synthesised by Jing et al. [15] in 1989. NG is made up of glassy nanoclusters connected by the glass-glass interfaces prepared via compacting glassy nanoparticles [16]. Fang et al. [17] investigated the deformation structure of Sc_75_Fe_25_ nanoglass and found that nanoglass could undergo remarkable plastic deformation, which was attributed to the effect of glass-glass interface. Sopu et al. [18] found that the glassy interfaces could play a precursor role in the formation of shear bands (SBs). Adibi et al. [19] investigated the effect of the glassy grain size and found that, with the decrease of the grain size, the deformation mode changes from a single SB to uniform superplastic flow. Softening towards the NG occurs during superplastic flow, so Sha et al. [20] proposed a bimodal grain size model, which could improve the strength without sacrificing super-plasticity. Besides the attractive mechanical properties, NG exhibits enhanced thermal stability compared with MG (Metallic glass) [21,22], attributed to the lower free energy state of NG compared with MG [23]. Based on the properties of nanoglass mentioned above, Gleiter [14] predicted that an age of new technologies based on non-crystalline materials would begin. Kushima et al. [24] found that the crystalline structure of ZnO nanowire electrode was transformed into NG during the first charge in the Li-ion nanobattery, which implies the necessity to investigate the mechanical properties of NG ceramics.

However, less progress in the mechanical properties of the NG ceramics could be found in the literature. In addition, it is difficult to obtain the information about the inhomogeneous structure of NGs in experiments, as X-ray or neutron-based techniques can only provide average structural information [23]. Molecular dynamics simulation (MD) can serve as an effective means to gain an insight into the microstructure in NG ceramics and their evolution during deformation. MD simulation has been widely adopted to investigate different kinds of mechanical response of various atomic structures, such as dislocations [25,26], nano-twin boundaries [27,28], and shear bands [13,29], etc. with the Large-scale Atomic/Molecular Massively Parallel Simulator (LAMMPS) algorithm [30]. 

In this work, we aim to confirm the effect and the influencing effects of nanoglass structure that may help to improve or achieve the super ductility of ceramics by changing its nanostructure without adding any other ingredients, for the purpose to promote the more extensive applications of ceramics in more fields. The mechanical property of nanoglass AlN (ng-AlN) is to be investigated using the MD simulations and compared with that of a-AlN, and the mechanisms of the super ductility of ng-AlN are to be uncovered. First, the short range order (SRO) and medium range order (MRO) nanostructures at the glassy interface are studied. Then, the effects of the size of the glassy particles in ng-AlN are explored. Finally, the temperature and strain rate effects during tension are investigated.

## 2. Computational Methods

Systematic simulations are performed with the Large-scale Atomic/Molecular Massively Parallel Simulator (LAMMPS) [30], which is open code and has the functions required for this research. The atomic interactions within the Al-N are modelled with the Vashishta potential [31], which has been widely used to investigate the deformation mechanism of AlN under different kinds of loading conditions [13,29,32,33]. The integration time step of 1 fs is adopted in all the simulations. First, to construct an amorphous AlN unit, a crystal unit (w-AlN) with the size of 49.76 (*x*) × 48.47 (*y*) × 49.80 (*z*) Å^3^ is built at first, then it is melted at 3500 K for 500 ps and cooled down to 10 K at a rate of 2 × 10^14^ K/s, at zero external pressure (NPT) ensemble with periodic boundary conditions (PBCs) applied in all dimensions [13]. Second, ng-AlN is constructed with the Voronoi tessellation method [34] using the “seeds” randomly distributed in the sample with the amorphous AlN unit as a source. The size of the cuboidal sample is 262 Å (*x*) × 51 Å (*y*) × 523 Å (*z*). In order to study the effect of grain size on the mechanical properties of nanoglass ceramic, 7, 27, and 436 seeds are distributed randomly and the corresponding average grain sizes are 8, 4, and 1 nm, respectively, as shown in Figure 1a. The grains are columnar with the generatrix along the *y* direction. To avoid an abnormal high stress concentration, the overlapped atoms at the glassy gains are deleted. Then, the conjugate gradient algorithm is adopted to minimize the energy of the structure. They are relaxed in isothermal-isobaric (NPT) ensemble by a Nose-Hoover thermostat at 10 K for 200 ps to release the residual stress with the PBCs applied to the three directions. For comparison, the amorphous AlN (a-AlN) is also generated by replication of the a-AlN unit [13], as mentioned above, with the same size of ng-AlN as illustrated in Figure 1a. It is then relaxed in an NPT ensemble for 200 ps at 10 K as well.

Uniaxial tension is applied along the *z*-direction for all the samples with a constant strain rate of 10^8^ s^−1^. The PBCs are used in *y*- and *z*-directions and the free surface condition towards the *x*-direction. The simulations are performed in an NVE (microcanonical) ensemble for 3000 ps at 10 K using a Langevin thermostat [35]. The temperature of 10 K is adopted to reduce the influence of the perturbation caused by the random vibration of the atoms. The atomic stress is calculated based on the virial stress [36]. Mises equivalent strain [37], *ε_equ_*, is employed to identify the plastic deformation of the sample, and the region where *ε_equ_* ≥ 0.2 is considered as a shear transformation zone (STZ) [13]. The software Ovito [38] is used to process and visualise the simulation results.

## 3. Results and Discussion

### 3.1. Structural of ng-AlN

ng-AlN has an amorphous structure containing a certain fraction of interfaces, which is different from the conventional a-AlN. To better understand the structure of ng-AlN, the radial distribution functions (RDFs) of ng-AlN with different grain sizes and that of a-AlN are analysed and shown in Figure 2a. RDF is defined as the average density of the atoms in a thin spherical shell, with a reference atom as its center, and *r* and *r* + *dr* as its inner and outer radii, respectively. RDF can be expressed as:(1)$$g(r)=\frac{N(r,r+dr)}{4\pi r^{2}dr\rho }$$
where *N*(*r*, *r* + *dr*) the number of atoms in the spherical shell, and *ρ* is the average density of the sample. Figure 2a respectively shows the RDF curves of a-AlN, ng-AlN_*d* = 8 nm_, ng-AlN_*d* = 4 nm_, and ng-AlN_*d* = 1 nm_ where it can be seen that they almost coincide with each other. The locations of the first peak, which are related to the bond length of Al-N, are nearly identical; however, the height of the first peak (which is also defined as “intensity”) corresponding to a-AlN is higher than those of ng-AlNs, as shown in the upper inset in Figure 2a, indicating that the average coordination number (CN) of a-AlN is larger than that of ng-AlN. The intensity decreases with the increase of the glassy interface in the ng-AlN samples, indicating that the average CN decreases with the increase of the glassy interface. The distributions of CN in a-AlN and ng-AlNs with different grain sizes are shown in Figure 2b. As we know, only three kinds of short range order (SRO) structures exist in a-AlN, i.e., threefold-coordinated, fourfold-coordinated, and fivefold-coordinated atoms [13,39]. In the ng-AlN, we do not find any other SRO structures different from that in a-AlN, as shown in Figure 2b. The fraction of five-coordinated atoms covers only about 0.5%, while three-coordinated atoms and four-coordinated atoms account for nearly 30% and 70% in the ng-AlN of different particle sizes and in a-AlN, respectively. With the increase of interface, the fraction of four-coordinated atoms decreases and three-coordinated atoms increases (Figure 2b), leading to a reduction of average CN. The second peaks in the RDF curves (Figure 2a) related to the bond lengths of both Al-Al and N-N correspond to the medium range order (MRO) structure. With the increase of interface, the position of the second peak shifts leftwards, indicating the decrease of the lengths of both Al-Al and N-N bonds. It has been shown that there are four kinds of MRO structures in a-AlN, i.e., fourfold ring, sixfold ring, eightfold ring, and tenfold ring, respectively [13]. In the RDF curves, the sequence of the heights of the second peaks is similar to that of the first peaks, indicating that the MRO structures in ng-AlN are relatively sparse compared with those in a-AlN, and become sparser with the increase of the interface. In order to further investigate the properties of the glassy interface, the Voronoi volume [40] is calculated for a-AlN and ng-AlNs with different grain sizes after relaxation, as shown in Figure 1b, where it can be found that the atomic volume at the glassy interface is larger than that in the intragranular part, which should be related to the decrease of the CN of ng-AlN.

### 3.2. Comparison between Mechanical Properties of ng-AlN and a-AlN

The stress strain (*σ* − *ε*) curves of the ng-AlNs with the grain sizes of *d* = 1 nm, 4 nm, and 8 nm, respectively, are shown in Figure 3, where the *σ* − *ε* curve of a-AlN is also provided for comparison. To better distinguish the elastic moduli of these samples, we calculate the bulk modulus, *B* and Young’s modulus, *Y*, of these samples with the elastic constants using the following relationships,
(2)$$B=\frac{C_{11}+2C_{12}}{3}$$
(3)$$Y=\frac{\left(C_{11}+2C_{12}\right)\left(C_{11}-C_{12}\right)}{C_{11}+C_{12}}.$$

The calculated *B* and *Y* of these samples shown in Table 1, where it can be seen that *B* and *Y* of ng-AlNs are smaller than those of a-AlN, and they decrease with the decrease of the grain size. Such kind of reduction in elastic property could be related to higher free volume at glassy interface, as shown in Figure 1b. Meanwhile, the cohesive energy decreases with the increase of grain size, which implies that the structure with larger grain size should be more stable. It means that the glassy interface has higher energy than the amorphous matrix. The ultimate strengths of ng-AlNs are relatively smaller than that of a-AlN and decrease with the decrease of grain size, as illustrated in Figure 3, which should be the result of the softer glass-glass interface, i.e., the more the glassy interface, the softer the ng-AlN.

With the increase of the glassy interface, the sample becomes more ductile, as shown in Figure 3. In order to investigate the deformation mechanism of the enhanced ductility with the decrease of the glassy grain size, the local shear strain is shown in Figure 4. Different from a-AlN, STZs are mainly activated in the glassy interface. *σ* reaches the ultimate strength at *ε* = 0.1204, 0.1208, 0.1334, and 0.1402, for a-AlN, ng-AlN_*d* = 8 nm_, ng-AlN_*d* = 4 nm_, and ng-AlN_*d* = 1 nm_, respectively, as shown in Figure 4, which could be ascribed to that the free volume at the glassy interface is larger than that in the matrix. The main shear band forms in a-AlN at *ε* = 0.180, as shown in Figure 4a. While in ng-AlN_*d* = 8 nm_ and ng-AlN_*d* = 4 nm_, STZs form at the glassy interface coalesce, propagate over the entire sample, and develop to shear bands (SBs) at *ε* = 0.180, as shown in Figure 4b,c. The SBs interact with each other in ng-AlNs due to the existence of glassy interfaces, leading to the slow drop of stress, as shown in Figure 3. The formation and propagation of SBs are driven by the elastic energy [18], and at the glassy interface the local energy is released, which may suppress the formation of mature shear band, accounting for that the width of the shear bands in ng-AlNs is smaller than that in a-AlN. Subsequently, voids initiate and grow in the SBs at *ε* = 0.1900, 0.2150, and 0.2177 in a-AlN, ng-AlN_*d* = 8 nm_, and ng-AlN_*d* = 4 nm_, as shown in Figure 4a–c, respectively. With the further increase of *ε*, the voids coalesce, giving rise to eventual destruction of the a-AlN, ng-AlN_*d* = 8 nm_ and ng-AlN_*d* = 4 nm_ sample at *ε* = 0.3. It is noteworthy that STZs emerge in the entire ng-AlN_*d* = 1 nm_ at *ε* = 0.18, as the result of high fraction of the glassy interface (Figure 4d). During the further loading, ng-AlN_*d* = 1 nm_ tends to deform uniformly without forming voids, exhibiting superplasticity, as can be seen in the *σ* − *ε* curves in Figure 3.

The initiation and propagation of STZs associated with temporal and spatial distributions are closely related to the deformation mode [19,41]. In ng-AlN_*d* = 8 nm_ and ng-AlN_*d* = 4 nm_, although STZs develop mainly at the glassy interfaces at first, they expand through glassy grain boundaries, forming SBs that dominate the plastic deformation and may suppress the formation of other STZs at the glassy interfaces, as shown in Figure 4b,c. When the grain size is reduced to 1 nm, besides the STZs developing initially in the glassy interface, other new STZs appear, which may lead to homogeneous deformation in the whole sample under extremely large strain. It indicates that the glassy grain size strongly affects deformation mode of nanoglass ceramics. The fractions of atoms with $\eta_{i}^{Mises}>0.2$ in a-AlN and ng-AlNs with different grain sizes during deformation process are shown in Figure 5. Overall, there are higher fraction of atoms at high shear strain level in ng-AlNs than that of a-AlN, due to the existence of glassy interface, and the fraction increases with the increase of glassy interface, due to the decrease of the grain size, as shown in Figure 5. It is noteworthy that the fraction of atoms at higher shear strain level in ng-AlN_*d* = 1 nm_ at *ε* = 0.3 is about 60%, indicating that most atoms participating in the shear deformation may lead to more uniform plastic deformation. Another interesting phenomenon is that in ng-AlN_*d* = 1 nm_ the fraction of atoms at high shear strain level before *ε* = 0.19 is smaller than that in the others, as shown in Figure 5. As the fraction of glassy interfaces in ng-AlN_*d* = 1 nm_ is much larger than that in the other ng-AlNs and glassy interface plays softeningrole, the stored elastic energy in ng-AlN_*d* = 1 nm_ should be less than that in the others, which could trigger relatively fewer STZs in the initial stage.

### 3.3. Effect of Temperature

As has been mentioned above, when the glassy grain size of ng-AlN is reduced to 1nm, it exhibits superplasticity. In this subsection, we further investigate the effect of temperature on the mechanical property of ng-AlN_*d* = 1 nm_ under tension, as shown in Figure 6. Before loading, the sample is relaxed at 150 K, 300 K, 450 K, and 600 K, respectively, to achieve the stable structure at the corresponding temperature. Similarly, we calculate the bulk and elastic moduli of these samples with the elastic constants at different temperatures, as shown in Table 2, where it can be seen that the temperature insignificantly influences the elastic property. In contrast, the ultimate strength is sensitive to the temperature, as can be seen in Figure 6. Figure 7 shows the effect of temperature (*T*) on the ultimate strength (*σ_b_*). It is obvious that *σ_b_* decreases with the increase of *T*. Johnson et al. [42] proposed a universal criterion for the temperature dependence on yield strength:(4)$$\sigma_{y}\left(T\right)=\sigma_{0}\left[1-C{\left(T/T_{g}\right)}^{2/3}\right]$$
where *σ*_0_ is a constant, *T_g_* the glass transition temperature of AlN (*T_g_* = 3070 K, predicted from MD simulations [31]) and the parameter $C\equiv {\left[g\left(k/\beta \right)\ln\left(\omega_{0}/\tau \dot{\varepsilon }\right)\right]}^{2/3}$ [43], which depends on the strain rate $\dot{\varepsilon }$. *g* measures the *T*-dependence of elastic moduli, *k* is the Boltzmann constant, and *ω*_0_ the frequency of shear waves of nm wavelength. These parameters and *β*, *τ* can be treated as constants [42]. The relationship between *σ_b_* and *T* can be well described by Equation (4) with *C* = 1.06, as shown in Figure 7, which can be ascribed to the fact that diffusive homogeneous flow may take place at a lower stress level with diffusive rearrangements at high temperature.

Next, we take 10 K, 300 K, and 600 K as the representatives to study the effect of temperature. It is noteworthy that, with the increase of the temperature, the number of three-coordinated atoms decreases, while that of four-coordinated atoms and five-coordinated atoms increases, as shown in Figure 8, which suggests that the free volume in the sample decreases with the increase of the temperature. The applied strains for ng-AlN_*d* = 1 nm_ to reach its ultimate strengths are *ε* = 0.1402, 0.1230, and 0.1184 corresponding to 10 K, 300 K, and 600 K, respectively, as shown in Figure 6, where it can be found that the applied strains decrease with the increase of the temperature, along with the emergence of STZs as illustrated in Figure 4d and Figure 9a,b. As has been mentioned above, although the free volume decreases with the increase of the temperature, which may weaken the activation of the STZs, the increase of temperature may help to overcome the activation barrier of critical shear event, and permit shear localisation to occur at low stress level [41]. It can be found that temperature does not change the homogeneous deformation behaviour of ng-AlN_*d* = 1 nm_, while the regions of STZs increase with the increase of the temperature at the same strain level, as comparing between Figure 4d and Figure 9a,b. For direct observation, the fractions of the atoms with $\eta_{i}^{Mises}>0.2$ in ng-AlN_*d* = 1 nm_ at different temperatures are shown in Figure 10, where it can be seen that the higher the temperature, the larger the fraction of atoms at higher shear strain level. It is worth to note that in ng-AlN_*d* = 1 nm_, the fraction of atoms with $\eta_{i}^{Mises}>0.2$ could even reach about 82% at *ε* = 0.3 in the case of 600 K.

### 3.4. Effect of Strain Rate

The effect of strain rate on the mechanical properties of ng-AlN_*d* = 1 nm_ under tension is shown in Figure 11. It can be found that the ultimate strength is positively correlated with strain rate. The variation of ultimate strength (*σ_b_*) against strain rate ($\dot{\varepsilon }$) is shown in Figure 12, where it is obvious that *σ_b_* increases with the increase of $\dot{\varepsilon }$. Symonds et al. [44] suggested a power law relationship for the dependence of yield strength, *σ_y_*, on $\dot{\varepsilon }$ as follows:(5)$$\sigma_{y}=A+B{\dot{\varepsilon }}^{C}$$
where *A*, *B*, and *C* are the fitting parameters. We extend Equation 5 to the description for the dependence of *σ_b_* on $\dot{\varepsilon }$, the parameters of which are fitted using the calculated results (Figure 12) as *A* = 5.68756, *B* = 1.727 × 10^−5^, *C* = 0.48901, and the fitting curve is also shown in Figure 12, where it can be seen that Equation (5) can well describe the dependence of *σ_b_* on $\dot{\varepsilon }$.

The ultimate stress (or the yield stress) is closely related to the activation of the STZs. The available free volume (the free volume which is mobile [45]) decreases with the increase of the $\dot{\varepsilon }$, ascribed to the less time for atoms to diffuse and for the free volume to be rearranged. The fewer available free volume implies that the activation of STZ needs higher stress, leading to strain rate hardening effect [46]. The fraction of atoms of $\eta_{i}^{Mises}>0.2$ in ng-AlN_*d* = 1 nm_ at different $\dot{\varepsilon }$ is shown in Figure 13a, where it can be seen that the fraction of STZs decreases with the increase of $\dot{\varepsilon }$ before *ε* = 0.243. It is rather remarkable that at *ε*_1_ = 0.243 the fraction of STZs in the sample under tension at $\dot{\varepsilon }=5\times 10^{8}s^{-1}$ exceeds that at $\dot{\varepsilon }=10^{8}s^{-1}$, and at *ε*_2_ = 0.265 the fraction of STZs in the sample under tension at $\dot{\varepsilon }=10^{9}s^{-1}$ exceeds that at $\dot{\varepsilon }=5\times 10^{8}s^{-1}$, as illustrated in Figure 13a. It can be attributed to that the heat induced by the formation of the first wave of STZs leads to the rapid activation of the next wave of STZs. The distributions of $\eta_{i}^{Mises}$ with STZs fraction of 0.2 (marked with the dashed line in Figure 13a) at $\dot{\varepsilon }=10^{8}s^{-1}$, $\dot{\varepsilon }=10^{9}s^{-1}$ and $\dot{\varepsilon }=10^{10}s^{-1}$ respectively, are shown in Figure 13b, where one can find that for the fixed fraction, the distribution of STZs becomes more uniform with the increase of $\dot{\varepsilon }$, attributed to that there is not sufficient time for atoms to diffuse and the free volume to be rearranged at a higher strain rate. At a higher strain rate, STZs would develop more uniformly, while at a lower strain rate, STZs would develop locally and then extend.

## 4. Conclusions

We investigated using MD simulations the mechanical properties of ng-AlNs with different sizes of glassy grains, which were subjected to uniaxial tension at different temperatures and strain rates. Some conclusions were drawn as follows:

1. The volume of the atoms at the glassy interface is larger than that in the intragranular part, due to the increase of three-coordinated atoms. The type of the SRO structures in ng-AlN is the same as that in a-AlN. The fraction of four-coordinated atoms decreases while that of three-coordinated atoms increases with the increase of interface. The MRO structure in ng-AlN is relatively sparse compared with that in a-AlN, and become sparser with the increase of interface.

2. The bulk and elastic moduli of ng-AlNs are smaller than those of a-AlN. They decrease with the decrease of the grain size, so do the ultimate strengths, which can be attributed to the weak glass-glass interface. STZs are mainly activated at the glassy interface in ng-AlN, which is different from that in a-AlN. When the size of glassy grain is reduced to 1 nm, ng-AlN would exhibit super-ductility and its deformation tends to be uniform without generating voids, which can be attributed to the uniform distribution of a great many of STZs generated.

3. Temperature insignificantly affects the elastic modulus of ng-AlN, but strongly affects the ultimate strength, which follows *T*^2/3^ scaling law for ng-AlN_*d* = 1 nm_. The uniform diffusive flow could take place at low stress, leading to lower strength. It was also found that the higher the temperature, the larger the fraction of STZs, which could even reach nearly 82% in ng-AlN_*d* = 1 nm_ at 600 K as *ε* = 0.3.

4. The ultimate strength of ng-AlN increases with the increase of strain rate, following a power law relationship. At a higher strain rate, there is less time for atoms to diffuse and for free volume to be rearranged, which may lead to the smaller available free volume inducing higher stress for the activation of STZ. Although at the initial deformation stage the number of STZs decreases with the increase of the strain rate, the number of STZs at a higher strain rate would exceed that at a lower strain rate.

## Figures and Tables

**Figure 1 nanomaterials-09-01535-f001:**
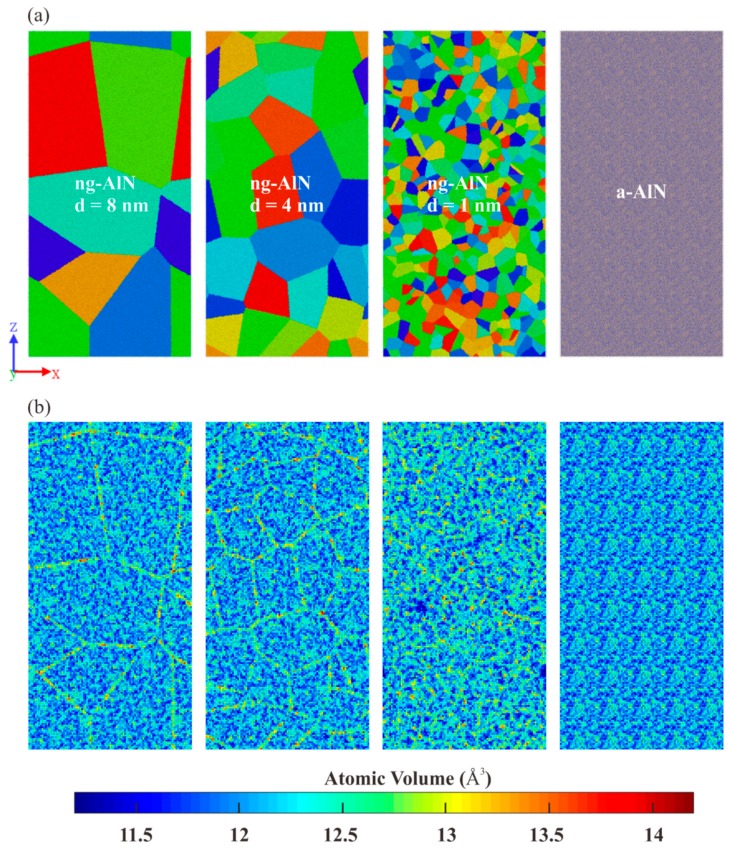
(**a**) Atomic configuration of ng-AlN with average grain size of 8 nm, 4 nm, and 1 nm, and a-AlN, respectively. Grains are coloured to highlight the architecture; (**b**) Distributions of atomic volume corresponding to these samples after relaxation, respectively.

**Figure 2 nanomaterials-09-01535-f002:**
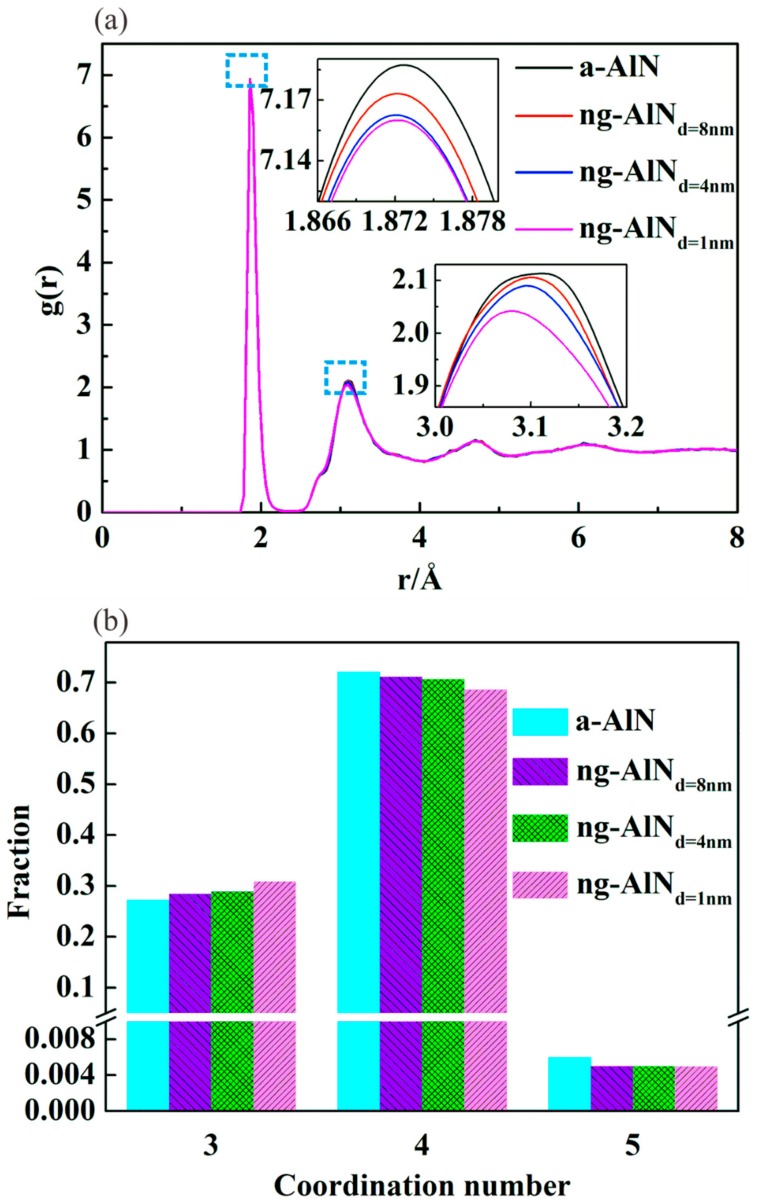
(**a**) Radial distribution functions (RDFs), (**b**) Coordination number of a-AlN, ng-AlN_*d* = 8 nm_, ng-AlN_*d* = 4 nm_, and ng-AlN_*d* = 1 nm_, respectively.

**Figure 3 nanomaterials-09-01535-f003:**
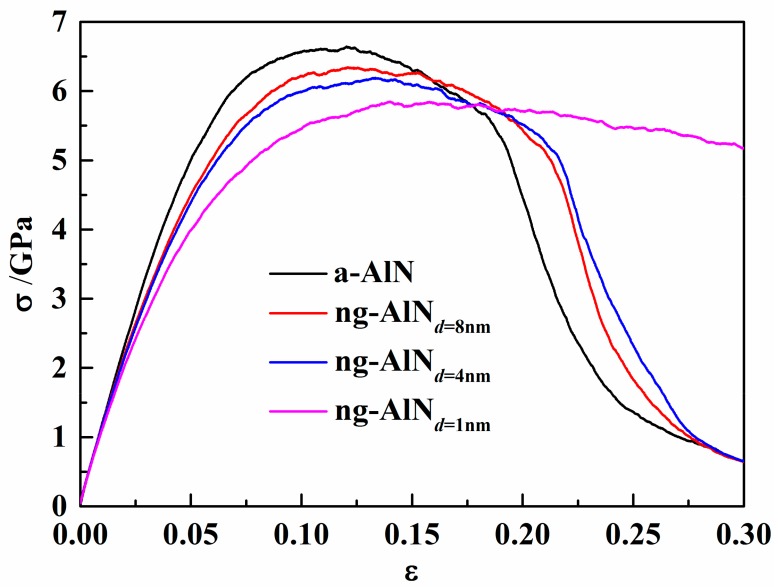
Tensile stress-strain (*σ* − *ε*) curves for a-AlN and ng-AlNs with average grain size of 8 nm, 4 nm, and 1 nm, respectively.

**Figure 4 nanomaterials-09-01535-f004:**
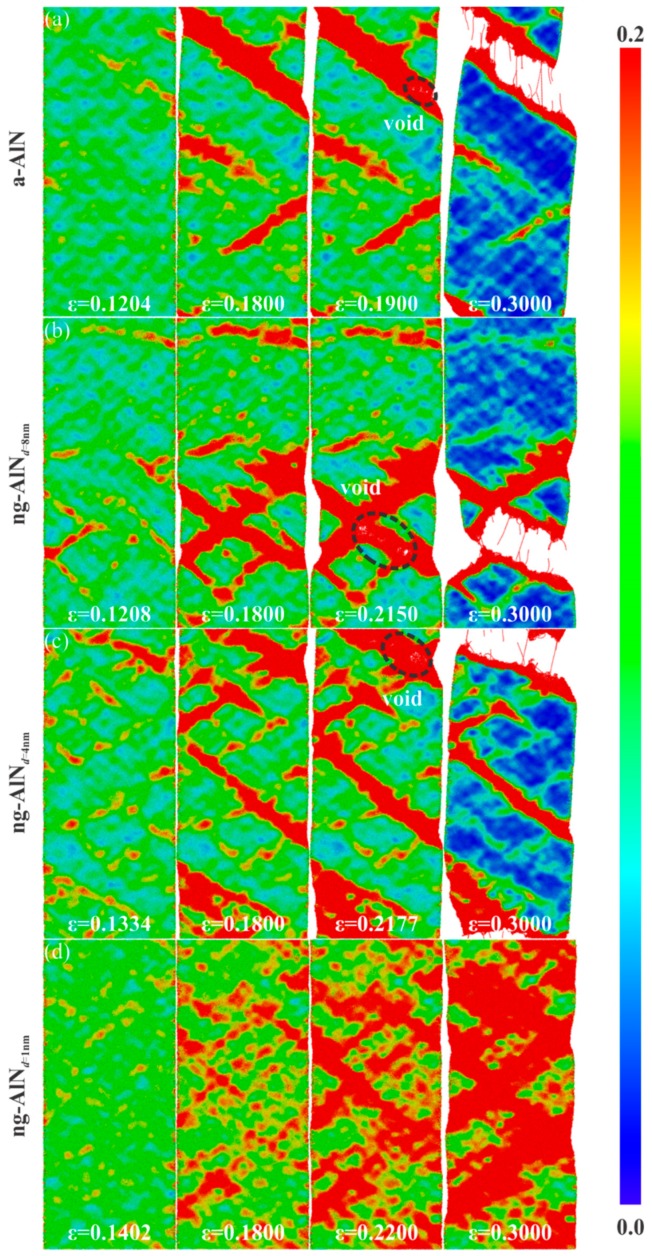
Evolution of von-Mises strain for (**a**) a-AlN, (**b**) ng-AlN_*d* = 8 nm_, (**c**) ng-AlN_*d* = 4 nm_, (**d**) ng-AlN_*d* = 1 nm_ at different strain levels.

**Figure 5 nanomaterials-09-01535-f005:**
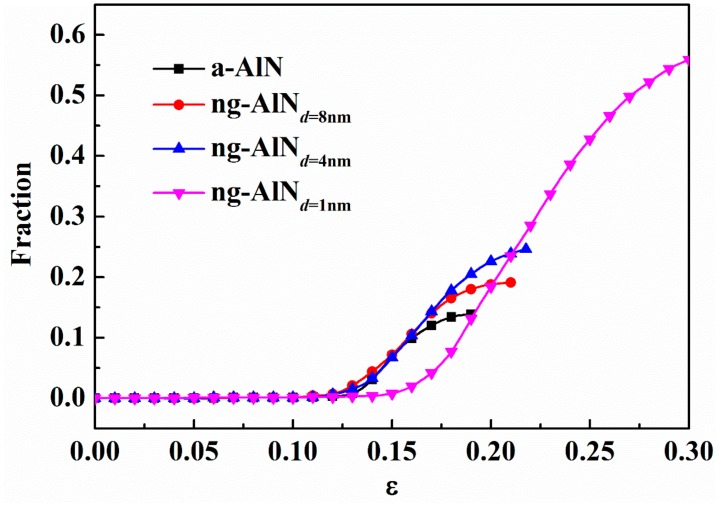
Fraction of atoms with relatively large atomic von Mises shear strain $\eta_{i}^{Mises}>0.2$ for a-AlN and ng-AlNs with different grain sizes at different strains.

**Figure 6 nanomaterials-09-01535-f006:**
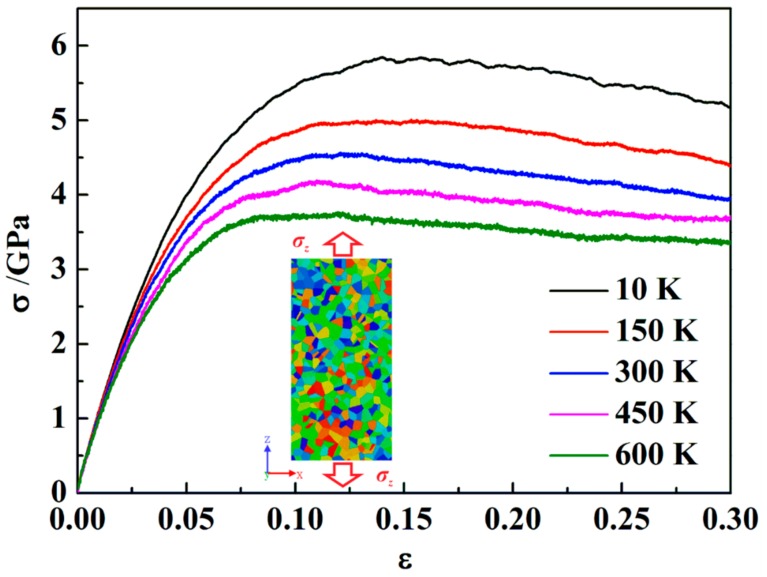
Tensile stress-strain (*σ* − *ε*) curves for ng-AlN_*d* = 1 nm_ under different temperatures.

**Figure 7 nanomaterials-09-01535-f007:**
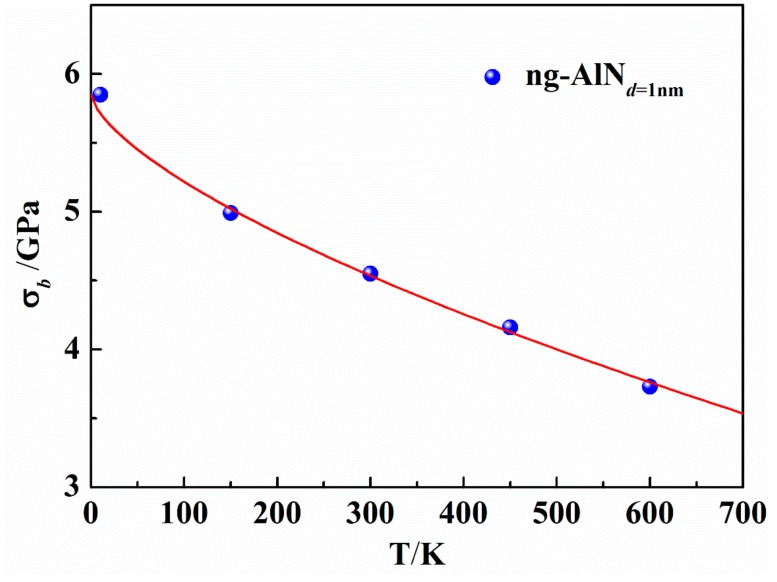
Ultimate strength as a function of temperature (*T*) for ng-AlN_*d* = 1 nm._

**Figure 8 nanomaterials-09-01535-f008:**
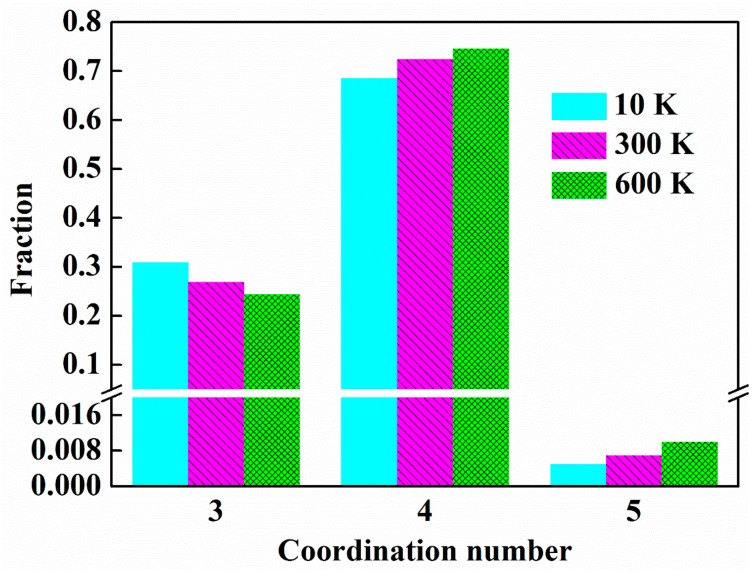
Radial distribution functions (RDFs) for ng-AlN_*d* = 1 nm_ at different temperatures—10 K, 300 K, 600 K.

**Figure 9 nanomaterials-09-01535-f009:**
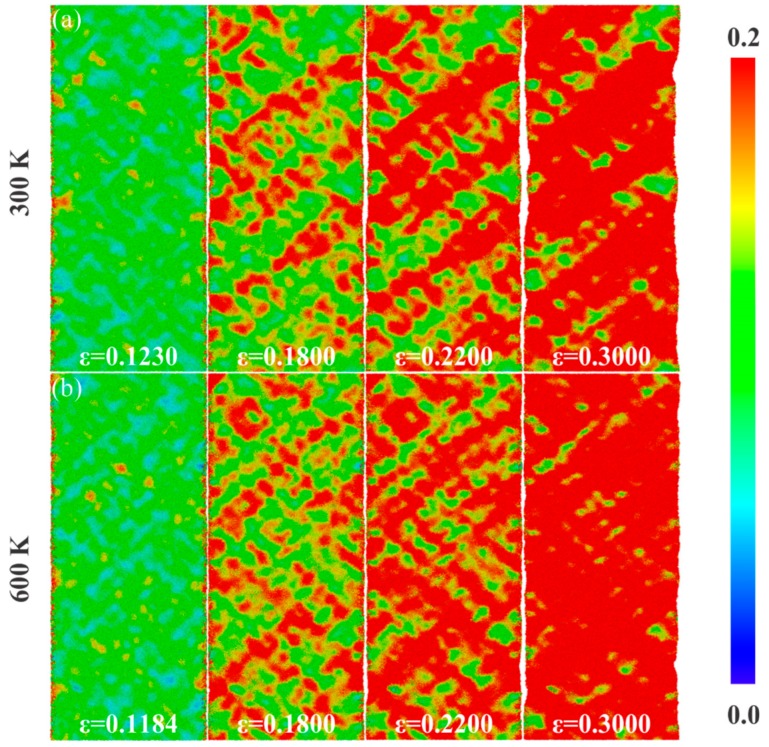
Evolution of von-Mises strain for ng-AlN_*d* = 1 nm_ at different strain levels at (**a**) 300 K and (**b**) 600 K, respectively.

**Figure 10 nanomaterials-09-01535-f010:**
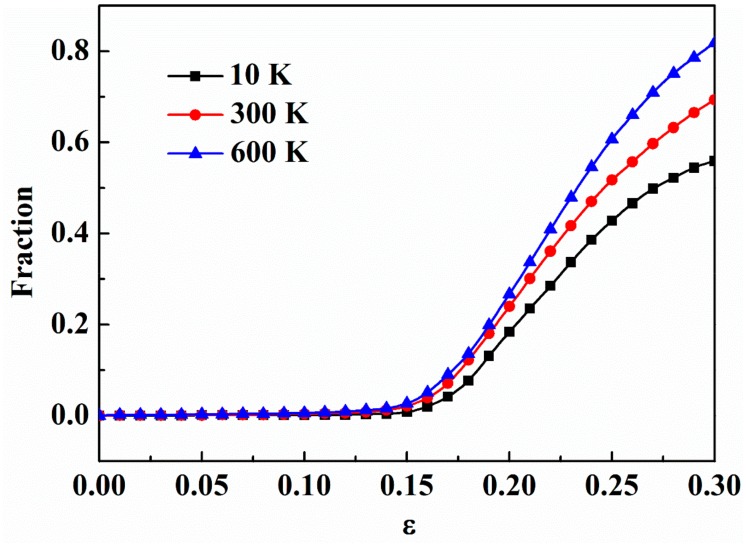
Fraction of atoms with relatively large atomic von Mises shear strain $\eta_{i}^{Mises}>0.2$ for ng-AlN_*d* = 1 nm_ at different temperatures with the variation of strain.

**Figure 11 nanomaterials-09-01535-f011:**
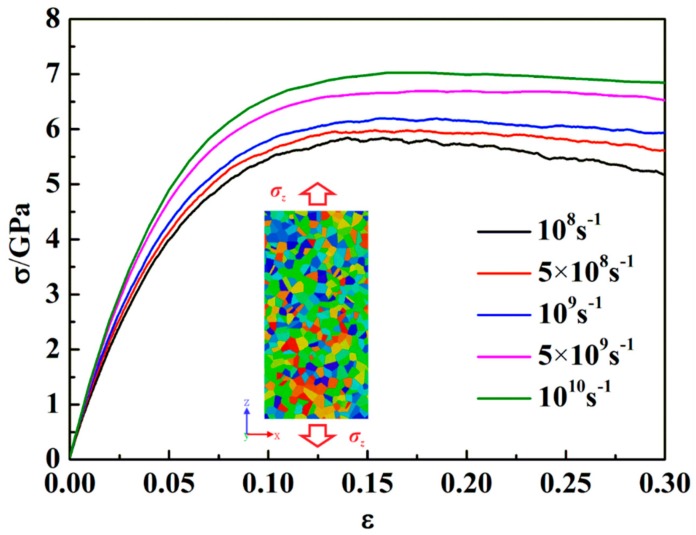
Tensile stress-strain (*σ* − *ε*) curves for ng-AlN_*d* = 1 nm_ at different strain rates.

**Figure 12 nanomaterials-09-01535-f012:**
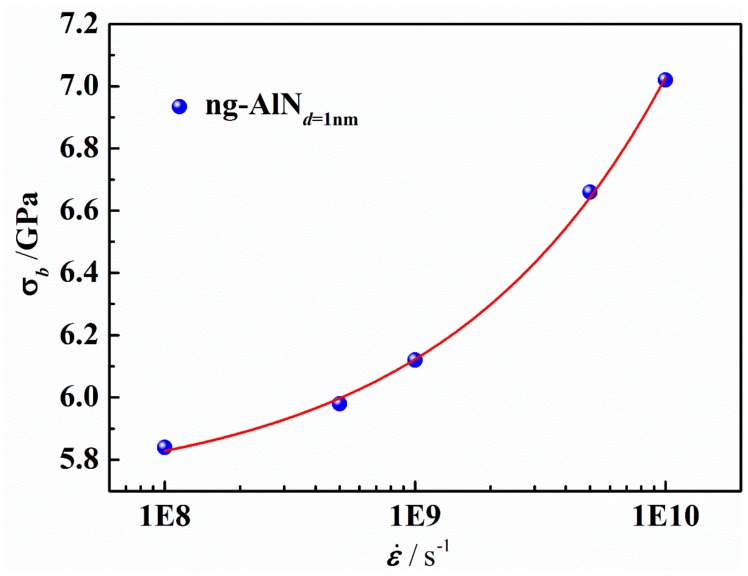
Ultimate strength as a function of strain rate ($\dot{\varepsilon }$) for ng-AlN_*d* = 1 nm._

**Figure 13 nanomaterials-09-01535-f013:**
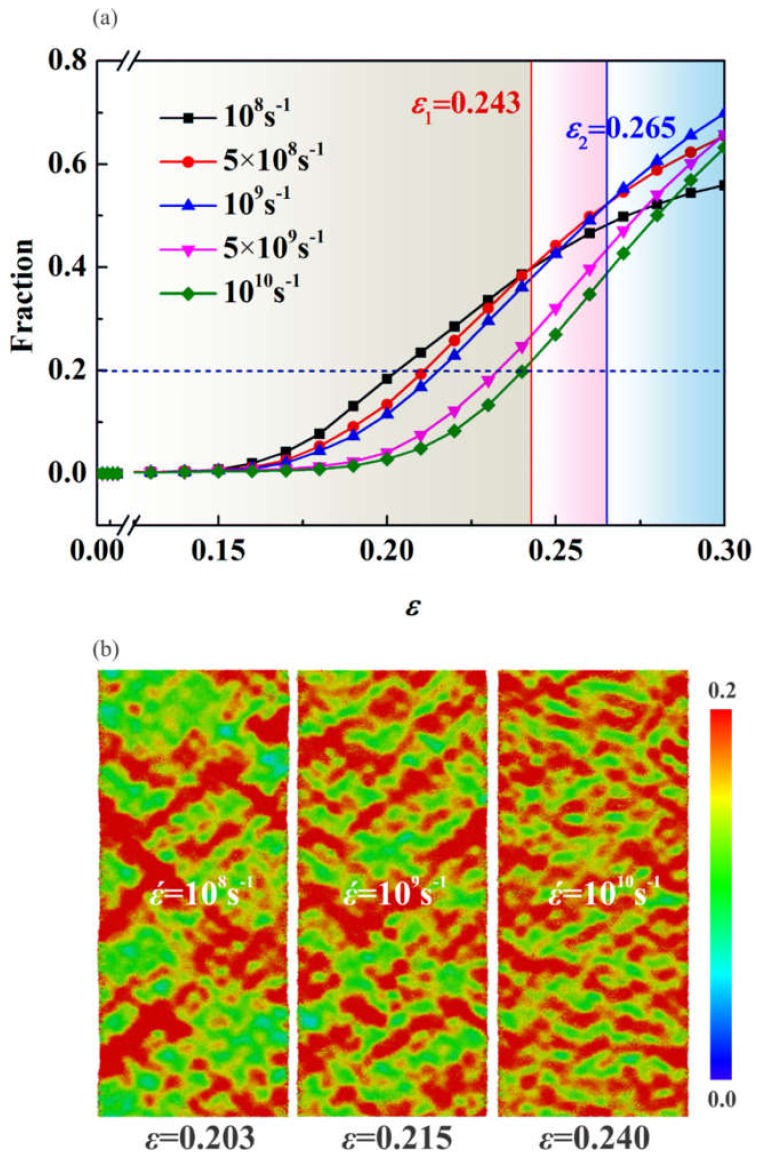
(**a**) Fraction of atoms with relatively large atomic von Mises shear strain $\eta_{i}^{Mises}>0.2$ for ng-AlN_*d* = 1 nm_ at different strain rates with the variation of strain; (**b**) Distributions of von-Mises strain for ng-AlN_*d* = 1 nm_ under strain rate of $\dot{\varepsilon }=10^{8}s^{-1},\;10^{9}s^{-1}\;\mathrm{and}\;10^{10}s^{-1}$ corresponding to strain *ε* = 0.203, 0.215, and 0.240, respectively.

**Table 1 nanomaterials-09-01535-t001:** Mechanical properties of a-AlN and ng-AlNs obtained with the Molecular dynamics simulation (MD) simulations. Notations: *Ec* (eV), cohesive energy; *B* (GPa), bulk modulus; *C*_11_ and *C*_12_ (GPa), elastic constant; *Y* (GPa), Young modulus.

	*Ec* (eV)	*B* (GPa)	*C*_11_ (GPa)	*C*_12_ (GPa)	*Y* (GPa)
a-AlN	−5.480	98.8	155.6	70.4	111.7
ng-AlN_*d* = 8 nm_	−5.474	96.6	153.0	68.4	110.7
ng-AlN_*d* = 4 nm_	−5.469	95.1	151.0	67.2	109.6
ng-AlN_*d* = 1 nm_	−5.454	90.9	145.6	63.6	106.9

**Table 2 nanomaterials-09-01535-t002:** Mechanical properties of ng-AlN_*d* = 1 nm_ at different temperature (10 K, 150 K, 300 K, 450 K, and 600 K) obtained with the MD simulations.

	*B* (GPa)	*C*_11_ (GPa)	*C*_12_ (GPa)	*Y* (GPa)
10 K	90.9	145.6	63.6	106.9
150 K	92.7	147.7	65.2	107.7
300 K	92.6	147.7	65.0	107.9
450 K	91.5	146.0	64.2	106.8
600 K	89.2	142.7	62.4	104.7

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
