# Peer review of "Super Ductility of Nanoglass Aluminium Nitride"

_nanomaterials, 2019, doi:10.3390/nano9111535_

Round 1

Reviewer 1 Report

The paper by Yinbo Zhao et al.. entitled “Super Ductility of Nanoglass Aluminium Nitride” reports results of their results of computer simulation about the mechanical properties of nanoglass Aluminum Nitride (ng-AlN). They investigated the grain size, temperature, and strain-rate dependence of mechanical properties of ng-AlN precisely by visualizing the real-space image of the distribution of the strain and analyzing them to clarify the mechanism. Their presentation is clear, and sound based on the objective result. The referee recommends this paper should be published in “Nanomaterials”. But before the publication, the referee would like to suggest the authors improve the manuscript considering two points listed below,

1)   In l. 257 on page 13, The authors wrote, “We extend Eq. (5) to the dependence of …”. The referee cannot understand how to extend Eq. (5) in their analysis. Please clarify it.

2)   In Figure 13 (a), epsilon_2 appears in the graph and is emphasized by colorful backgrounds. However, the referee cannot find any description of epsilon_2 in the main text as well as the caption of this figure. Please describe epsilon_2.

Also, please clarify a rather minor point,

a)     Some abbreviation is used without definition. For example, “MG” in l. 48 and “NVE” in line 90 are used without definition.

Reviewer 2 Report

The present study is about the mechanical properties of nanoglass aluminium nitrides with different sizes of glassy grains, which were subjected to uniaxial tension at different temperatures and strain rates.

Generally, the study is well conducted and written. However, there are some small remarks.

Computational Methods

There are different data names in the Legend for the Table 1. and in the Table 1.

Table 1. Mechanical properties of a-AlN and ng-AlNs obtained with the MD simulations. Notations: Ec (eV), cohesive energy; B (GPa), bulk modulus; Cij (GPa), elastic constant ...

But in the Table 1. the data are named as C11 (GPa) and C12 (GPa).

Results and discussion

Road 160: there are missing gaps: d=8nm, d=4nm.  It is not constant in the text: sometimes: d = 1 nm, 4 nm, and 8 nm (Road 138).

Road 158: misprint:could be ascribed

Road 163: the abbreviation STZs is not explained.

Reviewer 3 Report

The manuscript is devoted to aluminium nitride AlN in the form of nanoglass. The authors analyzed its mechanical properties by molecular dynamics simulations, when the tested mateials represented different glassy grain sizes.

Comments

1. The authors should remove the wrong phrase: "... a member of the group III transition metal nitrides ..." and in this place insert: "... a member of nitrides with metals from IIIA group ...". Aluminium is not placed among the transition metals in the d segment in Periodic Table (groups 3-12 or IIIB-IIB).

2. At the end of Introduction traditionally should be inserted aim of the study expressed laconically. The aim should be answer on question: what for the research was undertaken, what a hypothesis wanted the authors to test?

The manuscript needs minor revision.
